# Numerical Evaluation of Complex Capacitance Measurement Using Pulse Excitation in Electrical Capacitance Tomography

Damian Wanta *, Oliwia Makowiecka, Waldemar T. Smolik, Jacek Kryszyn, Grzegorz Domański, Mateusz Midura and Przemysław Wróblewski

Faculty of Electronics and Information Technology, Warsaw University of Technology, 00-665 Warsaw, Poland; oliwia.makowiecka.stud@pw.edu.pl (O.M.); w.smolik@ire.pw.edu.pl (W.T.S.); jacek.kryszyn@pw.edu.pl (J.K.); g.domanski@ire.pw.edu.pl (G.D.); m.midura@ire.pw.edu.pl (M.M.); p.wroblewski@ire.pw.edu.pl (P.W.)
* Correspondence: d.wanta@ire.pw.edu.pl; Tel.: +48-22-2347577

**Abstract:** Electrical capacitance tomography (ECT) is a technique of imaging the distribution of permittivity inside an object under test. Capacitance is measured between the electrodes surrounding the object, and the image is reconstructed from these data by solving the inverse problem. Although both sinusoidal excitation and pulse excitation are used in the sensing circuit, only the AC method is used to measure both components of complex capacitance. In this article, a novel method of complex capacitance measurement using pulse excitation is proposed for ECT. The real and imaginary components are calculated from digital samples of the integrator response. A pulse shape in the front-end circuit was analyzed using the Laplace transform. The numerical simulations of the electric field inside the imaging volume as well as simulations of a pulse excitation in the front-end circuit were performed. The calculation of real and imaginary components using digital samples of the output signal was verified. The permittivity and conductivity images reconstructed for the test object were presented. The method enables imaging of permittivity and conductivity spatial distributions using capacitively coupled electrodes and may be an alternative measurement method for ECT as well as for electrical impedance tomography.

**Keywords:** complex capacitance; admittance measurement method; electrical impedance tomography; capacitive coupling; pulse analysis technique

## 1. Introduction

Electrical capacitance tomography (ECT) is an imaging technique in which the spatial distribution of electric permittivity in a domain is reconstructed from capacitance values measured between the electrodes surrounding the domain [1].

Although ECT offers images with low spatial resolution, this technique found application in chemical engineering for process monitoring due to its high frame rate capability, installation flexibility, and relatively low cost [2]. A typical application is the imaging of multiphase flows or mixing processes [3–6]. ECT is mainly used to visualize flows with a non-conductive continuous phase. Electrical resistance (or resistivity) tomography (ERT) is used for imaging mixtures with a continuous conductive phase [7,8]. The possibility of adapting the hardware to measure both permittivity and conductivity and work in both cases would be a great advantage.

ECT can be potentially used in medicine, similarly to electrical impedance tomography (EIT) [9,10]. Possible medical applications are lung monitoring for accumulating fluid and heart function and blood flow monitoring [11]. To be an alternative to EIT, the ECT should measure both real and imaginary components due to the complex admittivity of biological tissues [12]. Unlike EIT, ECT uses capacitively coupled electrodes that are comfortable for the patient but challenging in real measurements. The main issue is to ensure a constant and equal coupling capacitance for all electrodes [13,14]. ECT can also be used for spatially resolved impedance measurement techniques applied to cell and particle research [15].

Various circuits for capacitance measurements in an ECT system were proposed. The alternating current (AC) method and charge–discharge method are standards in this technique [2], although many modifications are known [16–21]. Both sinusoidal excitation and pulse excitation are used in the sensing circuit in ECT systems; however, the methods with sinusoidal excitation have only been proposed to measure both components of complex capacitance.

Pulse excitation was studied in impedance measurement [22,23] and impedance spectroscopy [24]. Examples of impedance measurement with excitation in the form of a rectangular pulse can be found in [25]. In impedance spectroscopy, the use of pulse excitation of a short duration gives a shorter measurement time. Broadband test signals with a spectrum covering a wide frequency range are used together with the discrete Fourier transform to obtain spectral information about the test object [26]. The excitation signals can have various shapes [24], e.g., rectangular [27], Gaussian functions [28], chirp signals [29], sinc signals [30], and multi-sine signals [31]. Although the idea of application of pulse excitation in impedance measurement is known, it has not been explored in ECT so far.

In this paper, a novel method of complex capacitance measurement using pulse excitation is proposed for ECT. The real and imaginary components are calculated from the digital samples of the integrator's response to the rectangular pulse excitation. The advantage of the method is that it obtains a broadband response of the examined medium in one stimulation without the necessity to repeat it. The disadvantage of the method is the inability to obtain spectral information.

The proposed method uses a two-electrode measurement as opposed to the four-electrode measurement used in the EIT. The electrodes are capacitively coupled with the measured medium without ohmic contact. Similarly, two-electrode measurement is used in capacitively coupled impedance tomography (CCIT), which was proposed for medical or biological imaging [12,32], but sinusoidal excitation was used in that method.

Laplace analysis of the rectangular excitation in the front-end circuit was used to derive the analytical formulas for the components of complex capacitance. The numerical simulations of integrator response, including digital sampling, were performed to verify the validity of the method. The tomographic data were simulated, and the images were reconstructed to show the possibility of permittivity and conductivity imaging.

## 2. Materials and Methods

### 2.1. Admittance Measurement with Capacitive Coupling

The capacitance between electrodes is measured in ECT. Its value results from medium electric permittivity. When the medium is also conductive, both real and imaginary components of so-called complex capacitance are measured. The complex capacitance, defined as

$$C = C' - iC'', \tag{1}$$

results from complex permittivity of medium given by the following formula:

$$\varepsilon = \varepsilon' - i\frac{\sigma}{\omega}, \tag{2}$$

where $\varepsilon'$ is the electric permittivity of a tested medium, $\sigma$ is the conductivity, and $\omega$ is the angular frequency. The imaginary part of capacitance is directly related to conductance $G$.

$$C'' = \frac{G}{\omega} \tag{3}$$

Admittance of capacitance is defined by

$$Y(\omega) = i\omega C, \tag{4}$$

which for complex capacitance takes the form

$$Y(\omega) = G + i\omega C'. \tag{5}$$

The equivalent circuit model for a complex capacitance measurement is shown in Figure 1a, where $C_x$ and $G_x$ are the tested medium capacitance and conductance, respectively. $C_c$ is the total coupling capacitance of two electrodes. If $R_x = 1/G_x$ denote the resistance of the medium, the admittance of the circuit can be expressed by the following formula:

$$Y(\omega) = \frac{i\omega C_c(1 + i\omega R_x C_x)}{1 + i\omega R_x(C_x + C_c)}. \tag{6}$$

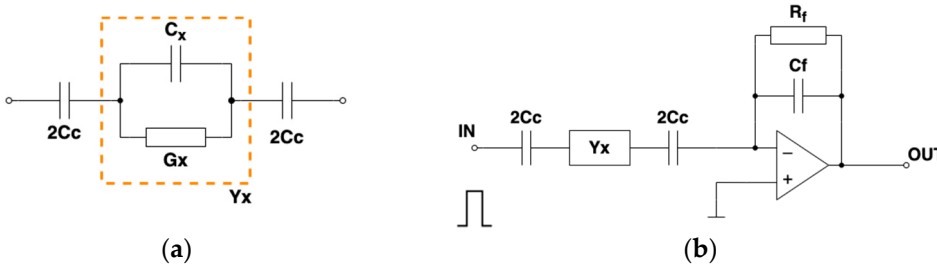

**Figure 1.** (**a**) Equivalent circuit for a model of complex capacitance measurement with capacitive coupling. (**b**) Integrator-based measurement circuit.

In the basic method of capacitance measurement, the voltage is applied to the measured capacitance, and the capacitance charging current is integrated. The charge $Q$ collected on the capacitor plate is proportional to the applied voltage $U$ and the capacitance value $C$. The circuit with the integrator, built using an operational amplifier with a negative feedback loop, is shown in Figure 1b. The integrator, also called a charge amplifier, produces the output voltage proportional to the charge transferred through the capacitor during the charging process. Resistor $R_f$ in the feedback loop is used to prevent saturation of the realistic integrator caused by the input bias current and leakage currents. We assumed that $R_f$ value is very high and does not influence the integration. Analog switches are used to connect/disconnect the measured capacitance to the pulse generator output and the input of the integrator in the tomographic hardware.

### 2.2. Pulse Shape Analysis in the Measurement Circuit

The integrator's response to the rectangular pulse excitation (Figure 2a) was calculated using Laplace transform for the model of measured admittance. The analysis was split into two parts as the Laplace transform of the rectangular pulse of duration $T$ can be separated into two transforms of the step functions.

$$\mathcal{L}(1\mathrm{I}(t_1) - 1\mathrm{I}(t_2 - T)) = \frac{1}{s}e^{-st_1} + \frac{1}{s}e^{-st_2}, \tag{7}$$

where $T = t_2 - t_1$. First, the rising edge of the pulse was analyzed. The step function with the amplitude $k$ and the rising edge at the time $t_1$ can be described by

$$u_{IN}(t) = k \cdot 1\mathrm{I}(t - t_1). \tag{8}$$

The signal representation in *s*-domain is given by

$$U_{IN}(s) = \frac{k}{s}e^{-st_1}. \tag{9}$$

The Laplace transform of the integrator input admittance $Y(s)$ is given by the following formula:

$$Y(s) = sC_c||(G_x + sC_x) = \frac{sC_c(G_x + sC_x)}{G_x + s(C_x + C_c)},$$  (10)

where $G_x + sC_x$ is a tested admittance and $C_c$ is a coupling capacitance. The current on the input impedance resulting from the step function in $s$-domain is given by

$$I(s) = U_{IN}(s)Y(s) = k\left[\frac{C_c(G_x + sC_x)}{G_x + s(C_x + C_c)}\right]e^{-st_1}.$$  (11)

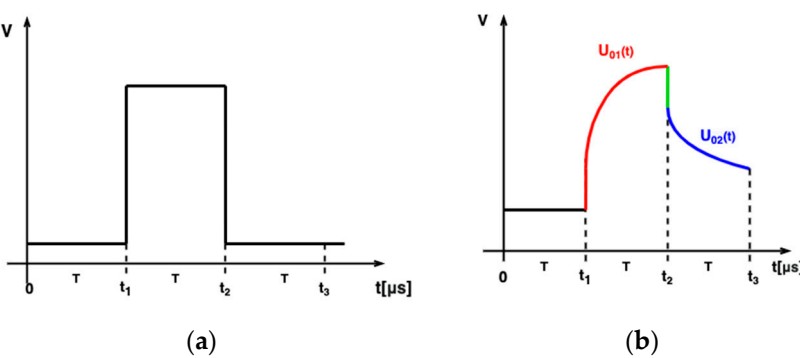

**Figure 2.** (**a**) Rectangular pulse excitation; (**b**) integrator's output.

Taking inverse Laplace's transform of the above equation, the current in function in time can be calculated. After substituting $R_x = 1/G_x$, the formula takes the following form:

$$i(t) = k\left[\frac{C_xC_c}{C_x + C_c}\delta(t - t_1) + \frac{C_c^2}{R_x(C_x + C_c)^2}e^{-\frac{t-t_1}{R_x[C_x+C_c]}}\right] \text{ for } t \in (t_1, t_2).$$  (12)

The response of the ideal integrator $u_{O1}$ with the gain $K$ to the step function can be derived by integrating the current (Figure 2b).

$$u_{O1}(t) = m\left[\frac{C_xC_c}{C_x + C_c} + \frac{C_c^2}{C_x + C_c}\left(1 - e^{-\frac{t-t_1}{R_x[C_x+C_c]}}\right)\right] \text{ for } t \in (t_1, t_2),$$  (13)

where $m = kK$.

Similarly, using the Laplace analysis, the trailing edge of the integrator response was analyzed (Figure 2b). The voltage at the integrator output after the falling edge of the exciting rectangular pulse, i.e., after the time $t_2$, is described by the following formula:

$$u_{O2}(t) = \left[u_{O1}(t_2) - m\frac{C_xC_c}{C_x + C_c}\right]e^{-\frac{t-t_2}{R_x[C_x+C_c]}} \text{ for } t > t_2.$$  (14)

The plots of the voltage response at the integrator output calculated using Equations (13) and (14) to rectangular pulse for different values of measured capacitance, conductance, and coupling capacitance are shown in Figures 3 and 4. The excitation pulse with the amplitude $k = 2.5$ V and duration $T = 14$ µS was used. The duration of the pulse corresponds to the duration of the excitation pulse used in our EVT4 tomographic device. In this device, a high voltage is used to measure non-conductive objects. A voltage of 2.5 V is sufficient to obtain a good signal in the case of conductive mixtures or biological tissue.

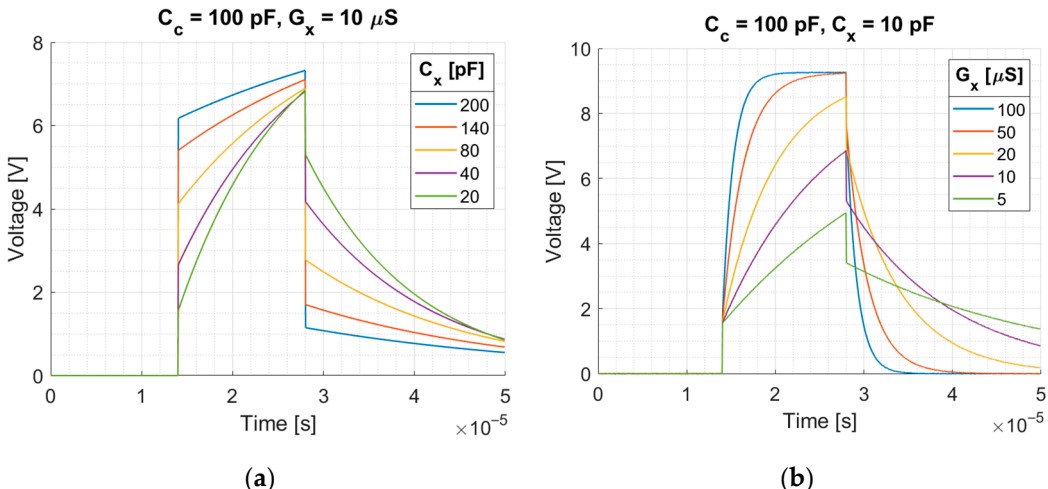

(a)

(b)

**Figure 3.** Integrator response to the rectangular pulse: (**a**) for different measured capacitance $C_x$ when $G_x = 10\ \mu S$, $C_c = 100$ pF; (**b**) for different measured conductance $G_x$ when $C_x = 10$ pF, $C_c = 100$ pF. Rectangular pulse: amplitude $k = 2.5$ V, duration $T = 14\ \mu S$.

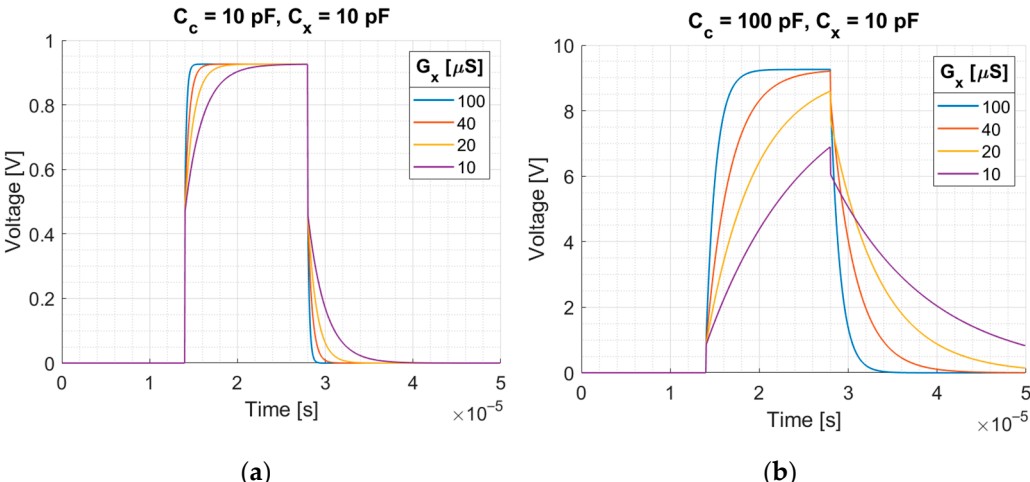

(a)

(b)

**Figure 4.** Integrator response to the rectangular pulse for different measured conductance $G_x$ and for different coupling capacitance: (**a**) $C_c = 10$ pF, (**b**) $C_c = 100$ pF, when $C_x = 10$ pF. Rectangular pulse: amplitude $k = 2.5$ V, duration $T = 14\ \mu S$.

The circuit simulations for different parameters were performed using Simulink (Figure 5). The rectangular pulse, with the parameters as above, was used. The plots of the voltage at the integrator output calculated analytically and obtained in the Simulink are presented in Figure 6 for a comparison. As an integrator built using an op-amp inverts the signal, the presented integrator output plots are inverted for ease of interpretation. Calculation based on analytical formulas and Simulink simulation gave the same result.

### 2.3. Admittance Component Estimation Using Digital Samples of Integrator Response

The proposed method of complex capacitance measurement is based on the analysis of integrator response shape. Let us assume that the pulse response of the integrator is sampled using an analog-to-digital converter. The sampling interval is divided into three phases of the same duration: before the pulse, during the pulse, and after the pulse

(Figure 7). At each phase, $N$ signal samples are collected. The samples collected in the first phase from the time $t_0 = 0$ to $t_1$ allow estimating the baseline

$$h_0 = \frac{1}{N} \sum_{n=0}^{N-1} u_n, \tag{15}$$

where $u_n$ is a value of the $n$-th sample. At the time $t_1$, the pulse front appears with a voltage surge of the value $h_1$ resulting from capacitance charging. The pulse ridge is sampled in the second phase between $t_1$ and $t_2$. Two parameters can be calculated from these samples. The area under the pulse ridge is calculated using the following formula:

$$A_1 = \frac{T}{N} \sum_{n=N}^{2N-1} u_n - h_0 T. \tag{16}$$

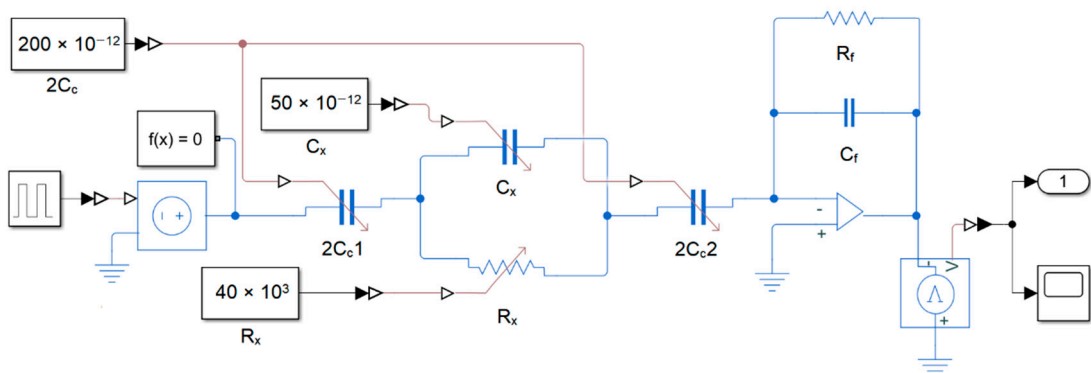

**Figure 5.** Circuit schema used in Simulink simulation.

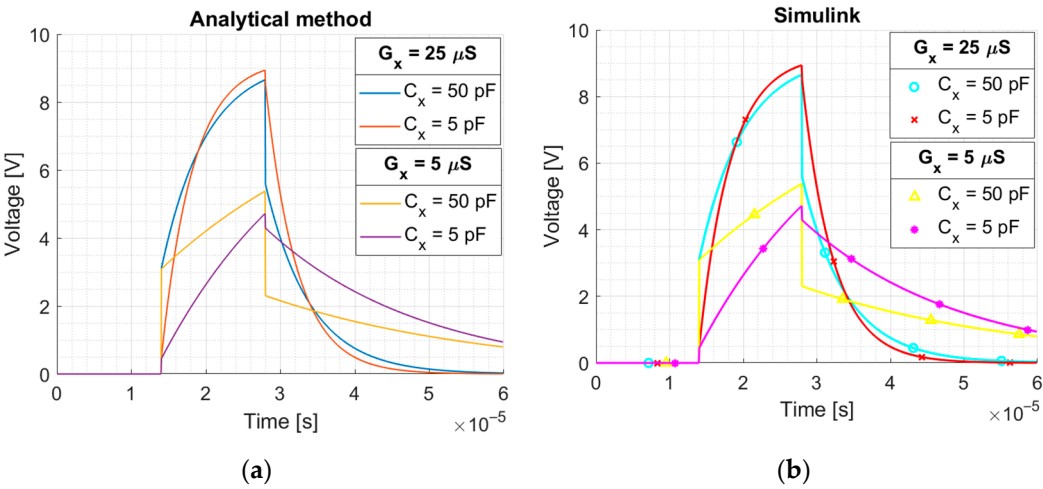

(**a**)             (**b**)

**Figure 6.** Integrator response to the rectangular pulse for different parameters: (**a**) calculated using derived formulas, (**b**) simulated using Simulink. Plots for $C_c$ = 100 pF. Rectangular pulse: amplitude $k$ = 2.5 V, duration $T$ = 14 μS.

The height of the pulse ridge $h_2$, resulting from charging through the medium resistance, is calculated using the following formula:

$$h_2 = \sum_{n=N}^{2N-1} (u_{n+1} - u_n). \tag{17}$$

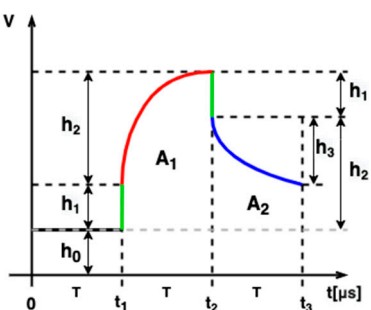

**Figure 7.** Integrator response pulse shape parameters.

The third sampling phase corresponds to the discharge phase after the excitation is turned off. The area under the pulse tail is calculated using the following formula:

$$A_2 = \frac{T}{N} \sum_{n=2N}^{3N-1} u_n - h_0 T. \tag{18}$$

The value by which the voltage drops in the time interval from $t_2$ to $t_3$ is determined using the following formula:

$$h_3 = - \sum_{i=2N}^{3N-1} (u_{n+1} - u_n). \tag{19}$$

$$h_0 = \frac{1}{N} \sum_{n=0}^{N-1} u_n, \tag{20}$$

where $u_n$ is a value of the $n$-th sample. At the time $t_1$, the pulse front appears with a voltage surge of the value $h_1$ resulting from capacitance charging. The pulse ridge is sampled in the second phase between $t_1$ and $t_2$. Two parameters can be calculated from these samples. The area under the pulse ridge is calculated using the following formula:

$$A_1 = \frac{T}{N} \sum_{n=N}^{2N-1} u_n - h_0 T. \tag{21}$$

The height of the pulse ridge $h_2$, resulting from charging through the medium resistance, is calculated using the following formula:

$$h_2 = \sum_{n=N}^{2N-1} (u_{n+1} - u_n). \tag{22}$$

The third sampling phase corresponds to the discharge phase after the excitation is turned off. The area under the pulse tail is calculated using the following formula:

$$A_2 = \frac{T}{N} \sum_{n=2N}^{3N-1} u_n - h_0 T. \tag{23}$$

The value by which the voltage drops in the time interval from $t_2$ to $t_3$ is determined using the following formula:

$$h_3 = - \sum_{i=2N}^{3N-1} (u_{n+1} - u_n). \tag{24}$$

The height of the voltage spike at the front of the pulse resulting from capacitance charging can be calculated using the Equation (13) by substituting $t = t_1$:

$$h_1 = m \frac{C_x C_c}{C_x + C_c}. \tag{25}$$

Similarly, setting $t = t_2$ in Equation (13) and subtracting $h_1$, the height of the output pulse ridge $h_2$ can be calculated.

$$h_2 = m \frac{C_c^2}{C_x + C_c} \left( 1 - e^{-\frac{T}{\tau}} \right), \tag{26}$$

where $\tau = R_x [C_x + C_c]$ and $T = t_2 - t_1$. The value by which the voltage drops in the discharge phase can be derived using the Equation (14) and is given by

$$h_3 = h_2 \left( 1 - e^{-\frac{T}{\tau}} \right). \tag{27}$$

The area under the pulse ridge can be calculated analytically by integrating Equation (13) in the range from $t_1$ to $t_2$:

$$A_1 = h_1 T + m \frac{C_c^2}{C_x + C_c} \left( T + \tau e^{-\frac{T}{\tau}} - \tau \right) \tag{28}$$

Similarly, the area under the pulse tail can be calculated analytically by calculating a definite integral of (14) in the range from $t_2$ to $t_3$:

$$A_2 = h_2 \tau \left( 1 - e^{-\frac{T}{\tau}} \right) \tag{29}$$

Using Formulas (26)–(29) and having the values of $h_2$, $h_3$, $A_1$, and $A_2$ calculated from the signal samples, formulas for the capacitance and conductivity of the medium can be derived. Since the voltage spike $h_1$, at the front of the output pulse, results from a capacitive divider, capacitance $C_x$ is expressed using the ratio of the height of the pulse ridge and the pulse tail. Conductance $G_x$ is calculated using the area under the pulse ridge and pulse tail.

$$C_x = m C_c^2 \frac{h_3}{h_2^2} - C_c \tag{30}$$

$$G_x = \frac{m A_2 C_c^2}{(m T C_c - A_1)^2} \tag{31}$$

In this derivation, it is assumed that the amplitude of the excitation square pulse, the integrator gain, and the value of the coupling capacitance are known. The analysis of the formulas shows that the method can be used only in a certain range of permittivity and conductivity values. This range depends on the value of the coupling capacitance, the assumed pulse duration, and the sampling frequency. The range of linearity of measurement of one component depends on the value of the other component. For both components, there are areas where the method is not sensitive to changes in the measured value. Figure 8 shows the dependence of the estimated value as a function of the true value for both capacitance and conductivity. The value of coupling capacitance was equal to 100 pF, the pulse duration was 14 µS, and the sampling frequency was 9 MHz. Under these conditions, the range over which the proposed method is sensitive to the change in permittivity and conductivity is respectively from 1 pF to 100 pF and from 10 µS to 1 mS.

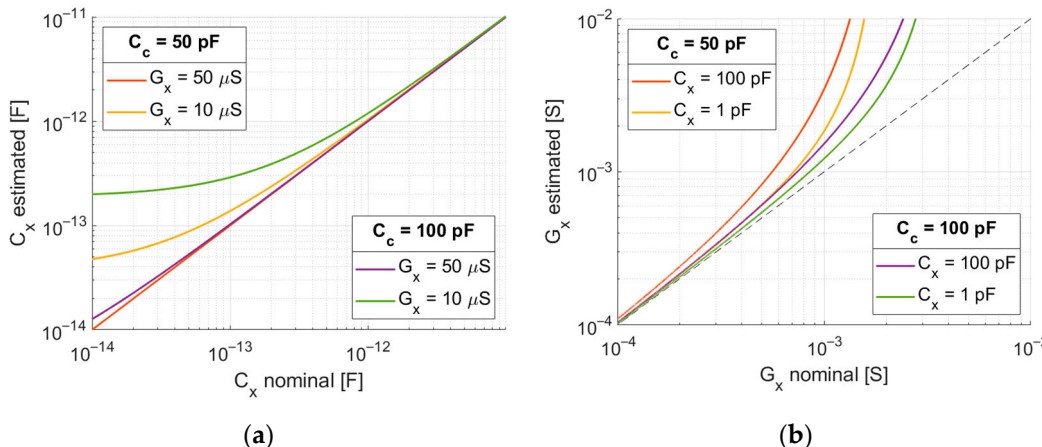

**Figure 8.** (**a**) Estimated value of capacitance as a function of nominal value for two different conductivity values. (**b**) Estimated value of conductivity as a function of real value for two different capacitance values. Coupling capacitance: $C_c = 50$ pF and 100 pF. Rectangular pulse: amplitude $k = 2.5$ V, duration $T = 14$ μS. ADC sampling frequency $f = 9$ MHz.

The proposed method requires analyzing the values of parameters determined from samples to protect against extreme values outside the assumed range. If any of the values of $h_2$, $h_3$, $A_1$, or $A_2$, particularly $A_2$ or $h_2$, is lower than the assumed level, the result must be rejected. We assume here that the values of capacitance and conductivity are within the assumed range, and we omit the analysis of the extreme cases.

The numerical simulations were performed to evaluate the method of admittance component estimation using the samples of the integrator's response. The white additive Gaussian noise was used in the generation of digital samples. The simulation of the integrator's response for given values of parameters was repeated several times to obtain different realizations of the random process. The mean value and standard deviation were calculated for conductance and capacitance. The analysis was performed in a selected range of capacitance and conductance values and for selected values of coupling capacitance. The measurement uncertainty, relative error, linearity, and sensitivity were calculated in a selected measurement range. Standard uncertainty represents a measure of the variation of a variable from its average. The relative standard uncertainty is defined as

$$u_{rs} = \frac{\sigma}{y_{avg}} \times 100\% \tag{32}$$

where $y_{avg}$ is a mean value and $\sigma$ is a standard deviation. The accuracy of measurement describes a closeness of the measurement value to the true value and can be expressed by relative error, defined as

$$e_r = \frac{|y - y_{true}|}{y_{true}} \times 100\% \tag{33}$$

where $y$ is a measured value and $y_{true}$ is a true value. The linearity error is defined as the deviation of a function from a linear relationship and is given by the following formula:

$$e_l = \frac{|\Delta y_{max}|}{x_{max} - x_{min}} \times 100\%, \tag{34}$$

where $\Delta y_{max}$ is the maximum difference between the measured value and the fitted line; $x_{min}$ and $x_{max}$ are the lower and upper range limits, respectively [33]. The sensitivity of the measuring device can be defined as the derivative of the response curve using an approximation:

$$s = \frac{\Delta y_{est}}{\Delta y_{nom}}, \tag{35}$$

where $\Delta y_{est}$ is the change in instrument response and $\Delta y_{nom}$ is a change in the measured quantity.

### 2.4. Numerical Simulations of Tomographic Measurements

Numerical simulations of two-dimensional (2D) tomographic measurements were carried out. In 2D ECT, a cross-section of the volume of interest is obtained from measurements carried out using one ring of electrodes.

To simulate tomographic projections, it is necessary to solve a forward problem in which the electric field and the so-called sensitivity matrix are determined. Knowing the sensitivity matrix, it is possible to determine the capacitance value between the electrodes for the assumed spatial distribution of complex permittivity.

In a real application, the response of the medium to the broadband excitation includes frequencies from a certain band. For some materials, electric parameters significantly vary in function of frequency. For example, biological tissues show a variable response over the frequency range from a few hertz to several megahertz. In the case of broadband excitation, the realistic simulation of tomographic measurement requires the integration of results over a wide frequency range. The frequency range depends on the shape of the used signal [34]. When a rectangular pulse is used, the spectrum is composed of odd frequencies and is limited to a certain range. The lower limit of this range results from the pulse duration, and the upper limit results from the rise time of the excitation pulse edge in the real measuring system. To reduce the complexity of the experiments, we assumed that complex permittivity was constant in the function of frequency. This simplified model, although not very realistic, remains valid for the verification of the proposed measurement method.

Solving a forward problem allows obtaining complex capacitances of the object without taking into account the parameters of the measuring electrodes and connecting leads. Such data cannot be observed in a real system. The data generated in the forward problem solver served as the reference data in the experiment. The second dataset was generated using the front-end circuit model and the proposed method of processing digital samples. The first dataset represented the ideal case, whereas the second represented the proposed measurement method.

#### 2.4.1. Test Object

The admittance of biological tissues is a complex quantity combining conductivity and capacitance. Thus, the 2D numerical model of the human thorax was used for the experiments (Figure 9). We defined the elements of the numerical phantom using standard geometrical primitives: ellipses and rectangles. As in other articles, the phantom geometry is based on a computed tomography cross-section of the chest [35]. The geometrical and electrical parameters of the phantom elements are given in Table 1. In the simulations, we assumed that the complex permittivity was constant within the frequency range, and we used the values specified for 500 kHz.

**Table 1.** Parameters of thorax numerical phantom (Figure 9) for 500 kHz. Permittivity and conductivity values are taken from [36].

| Tissue | Object | Position xc, yc [mm] | Size a, b [mm] | Tilt [deg] | $\varepsilon_r$ | $\sigma\left[\frac{mS}{m}\right]$ |
|---|---|---|---|---|---|---|
| left lung | LL | 59.0, −7.8 | 43.4, 71.0 | 70.9 | | |
| right lung | RL1 | −64.4, 0.6 | 54.4, 77.5 | 95.4 | 1029 | 123 |
| | RL2 | −53.6, 68.6 | 9.8, 33.2 | 176.44 | | |
| heart | H | 14.2, 38.2 | 46.3, 39.4 | 21.3 | 3260 | 281 |
| spine | S | −6.9, −47.6 | 11.4, 11.4 | 90 | 175 | 22.2 |
| aorta | A | −0.9, −23.7 | 9.9, 9.7 | 90 | 3260 | 281 |
| body | LBO | 65.0, 2.7 | 67.3, 98.22 | 90 | | |
| | RBO | −71.8, 4.7 | 69.7, 100.2 | 90 | 56.8 | 43.8 |
| | BR | 0, 0 | 143.6, 191.1 | 90 | | |

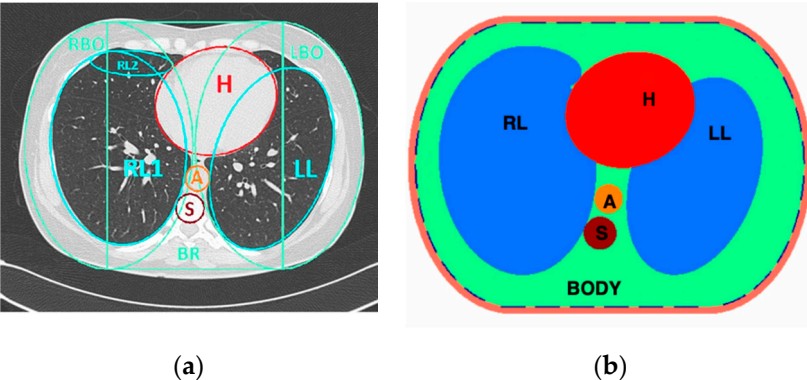

**(a)**                                **(b)**

**Figure 9.** (**a**) Computed tomography cross-section of the human chest with marked regions. (**b**) The geometry of the numerical phantom of the human thorax. H—heart, A—aorta, S—spine. Thirty-two electrodes used in the measurement and the outer screen are shown.

### 2.4.2. Forward Problem

ECTsim toolbox for Matlab developed in our laboratory was used for forward problem solving [37]. A finite volume method (FVM) and non-uniform square mesh are used to calculate electric field distribution in this toolbox [38].

The geometrical model of a test object was defined using geometrical elements and algebraic operations on these elements. Thirty-two measurement electrodes surrounding the object were added to the model (Figure 9). The size of the electrodes was 15 mm in width and 60 mm in height. The electric parameters given in Table 1 were used for the model elements. In the 2D simulation, it is assumed that the geometry and electrical properties are not changing in the $Z$-axis direction. Integration over a range corresponding to the height of the electrode in the $Z$-axis is performed to calculate the sensitivity or capacitance value for the electrode pair.

When using a multi-channel device, a measuring strategy is possible in which all but one selected electrode are simultaneously measuring electrodes. These electrodes measure current. The selected electrode is used to force voltage in the tested domain. During tomographic scanning, each electrode becomes a stimulating electrode, which gives $N(N-1)$ measurements, where $N$ is the number of electrodes. In fact, only $N(N-1)/2$ measurements are linearly independent since one electrode pair is measured twice in this sequence.

The potentials on the stimulating and measuring electrodes constitute the boundary conditions for the electric potential equation. The potential was equal to 2.5 V for the excitation electrode and 0 V for the measuring electrodes. The current measuring electrodes are at zero potential due to the connection to the virtual ground of the op-amp.

The electric field for all voltage excitations, the sensitivity matrix, and capacitance between electrode pairs were calculated using the toolbox (Figure 10). As the electric parameters of material, i.e., conductivity and permittivity, were defined as complex values, the complex electric field and complex sensitivity maps for electrode pairs were calculated [39]. The inverse problem was solved for a frequency equal to 500 kHz.

### 2.4.3. Simulations of the Measurements Using Pulse Excitation

Having the complex capacitance values determined by solving a forward problem, it is possible to simulate the integrators' responses during tomographic scanning. The model of measurements with capacitive coupling was used. Additive Gaussian noise was added to the generated digital samples. On the basis of the signal samples, the capacitance values for all electrode pairs were determined using Formulas (30) and (31).

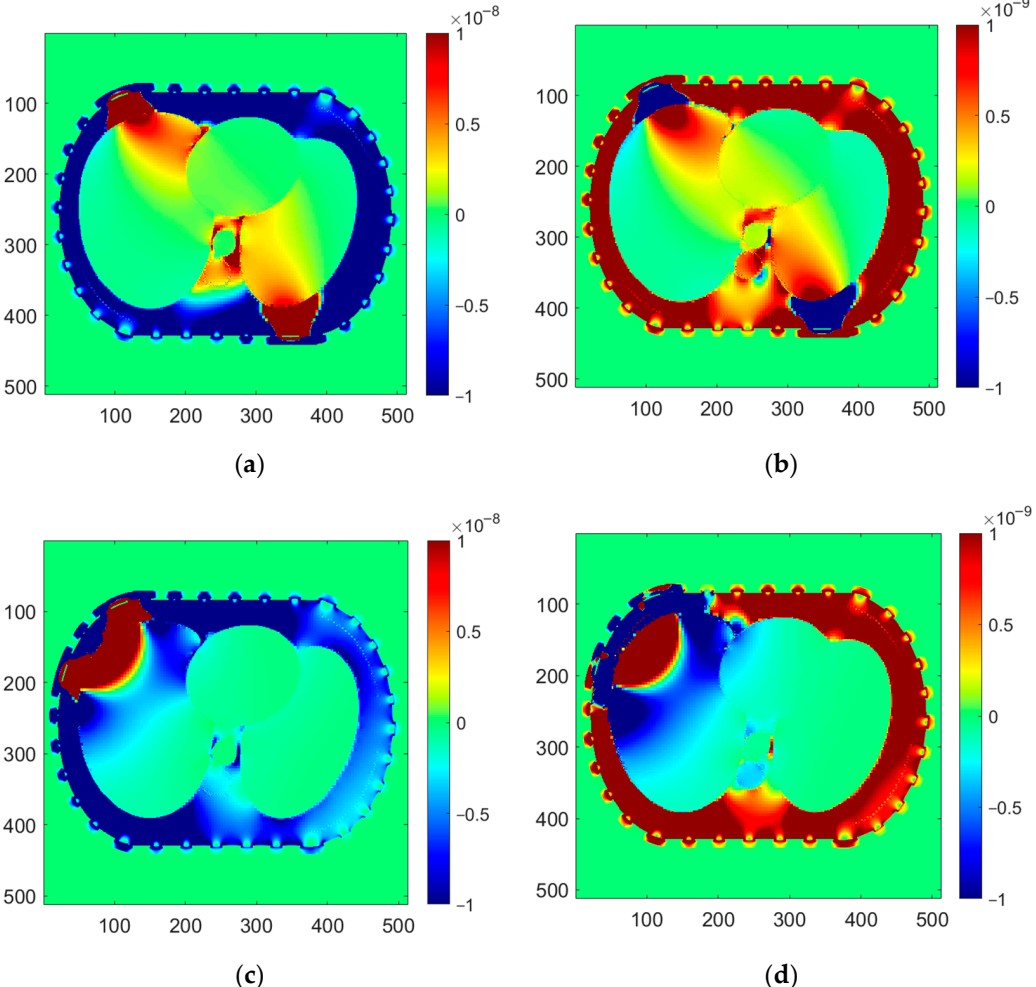

**Figure 10.** Sensitivity maps for selected pairs of electrodes for known permittivity and conductivity distribution: (**a**,**b**) opposite electrodes (1–16); (**c**,**d**) near electrodes (1–4); (**a**,**c**) maps for capacitance measurement; (**b**,**d**) maps for conductance measurement. The so-called "windowing" is applied to increase image contrast and show sensitivity distribution in the center of the thorax phantom.

*2.5. Image Reconstruction*

The reconstruction can be performed with and without the separation of two components. In the first case, real parts of complex voltage data are used to reconstruct conductivity images and their imaginary parts to reconstruct permittivity images. Since this approach neglects the mutual influence of conductivity and permittivity values on the electric field distribution in the examined space, we chose the second approach, in which the reconstruction algorithm is performed on complex measurements of capacitance and reconstructs complex permittivity. Reconstruction in a complex domain becomes important when $\varepsilon$ values increase to be comparable with $\sigma/\omega$, for example, for biological tissues at a frequency above 1 kHz.

The inverse problem was solved using a modified Levenberg–Marquardt (LM) algorithm. LM algorithm is an iterative algorithm in which, in each step, the sensitivity matrix is recalculated using a current estimate of the solution. To reduce the computational cost, the Jacobian matrix was computed only a few times every certain number of iterations. The value of the regularization parameter was selected experimentally.

## 3. Results

### 3.1. Admittance Estimation Using Simulated Integrator Response

The numerical simulations were carried out to evaluate the uncertainty of the admittance component estimation using the proposed method. The measured admittance was excited using the rectangular pulse whose amplitude was 2.5 V and duration was 14 µS. The range of integrator's output voltage was <0; −10 V>. The output signal was sampled with a frequency equal to 9 MHz. White Gaussian noise was added to the samples of the integrator's output with the magnitude 5 mV root-mean-squared (rms). In the experiment, the values of the parameters were selected that correspond to the values of the signals present in the tomographic device designed by our group. This allowed us to numerically verify the proposed method in the context of its potential implementation in our hardware.

The examples of noisy integrator output are shown in Figure 11. The values of conductance and capacitance were calculated from the samples using the Formulas (30) and (31). The simulations were repeated for the value of conductance in the range from 10 µS to 1 mS and the value of capacitance in the range from 1 pF to 100 pF.

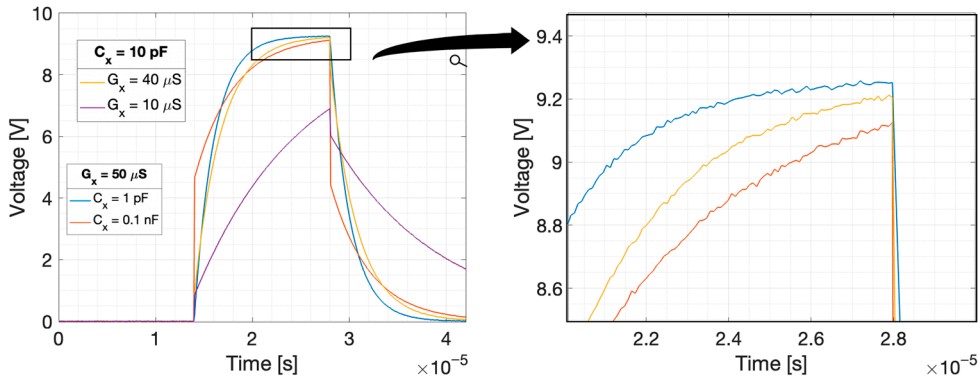

**Figure 11.** Integrator's response at the noise level of the magnitude 5 mV rms: for various measured capacitance $C_x$ and conductance $G_x$ at the value of coupling capacitance $C_c = 100$ pF.

The estimation quality parameters for the proposed measurement method were determined, assuming the measurement range from 1 pF to 100 pF for capacitance and from 10 µS to 1 mS for conductance. The circuit response curves for capacitance and conductance are presented in Figures 12 and 13. The response curve is a plot of an estimated versus a nominal value. The estimated value is the mean from 100 measurements. The standard uncertainty bars are plotted with the width corresponding to three-sigma. The relative error and sensitivity for three selected values from the measurement range are given in Tables 2–5. The linearity error for capacitance measurement is 0.24% for conductance $G_x = 0.01$ mS and 0.02% for conductance $G_x = 1$ mS (Figure 12). The linearity error for conductance measurement is 16.80% for capacitance $C_x = 1$ pF and 33.56% for capacitance $C_x = 100$ pF (Figure 13). The precision for conductance measurement is better, whereas the linearity is better for capacitance measurements.

**Table 2.** Capacitance estimation parameters for $G_x = 10$ µS, $C_c = 100$ pF, and 5 mV rms noise.

| Nominal $C_x$ [PF] | Estimated $C_x$ [PF] | Relative Error [%] | Uncertainty [%] | Sensitivity |
|---|---|---|---|---|
| 1 | 1.23 | 23.09 | 19.70 | 0.97 |
| 10 | 10.25 | 2.53 | 3.08 | 0.97 |
| 100 | 100.67 | 0.66 | 1.88 | 1.28 |

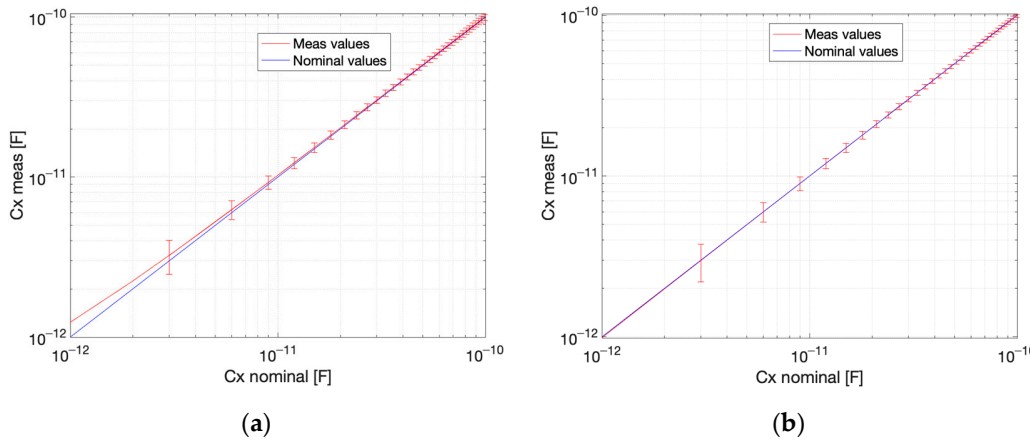

(**a**)                                              (**b**)

**Figure 12.** Estimated capacitance value vs. nominal value: (**a**) conductance $G_x = 0.01$ mS, (**b**) conductance $G_x = 1$ mS. Coupling capacitance $C_c = 100$ pF. Uncertainty bars represent the 3-sigma range.

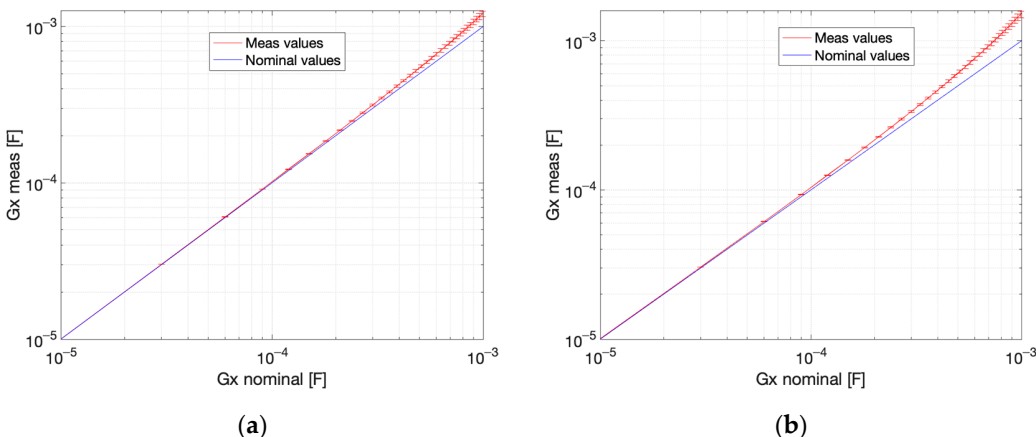

(**a**)                                              (**b**)

**Figure 13.** Estimated conductance value vs. nominal value: (**a**) capacitance $C_x = 1$ pF, (**b**) capacitance $C_x = 100$ pF. Coupling capacitance $C_c = 100$ pF. Uncertainty bars represent the 3-sigma range.

**Table 3.** Capacitance estimation parameters for $G_x = 1$ mS, $C_c = 100$ pF, and 5 mV rms noise.

| Nominal $C_x$ [PF] | Estimated $C_x$ [PF] | Relative Error [%] | Uncertainty [%] | Sensitivity |
|---|---|---|---|---|
| 1 | 0.96 | 3.16 | 27.00 | 1.04 |
| 10 | 10.00 | 0.01 | 2.72 | 1.00 |
| 100 | 99.98 | 0.01 | 0.85 | 1.00 |

**Table 4.** Conductance estimation parameters for $C_x = 1$ pF, $C_c = 100$ pF, and 5 mV rms noise.

| Nominal $G_x$ [PF] | Estimated $G_x$ [PF] | Relative Error [%] | Uncertainty [%] | Sensitivity |
|---|---|---|---|---|
| 0.01 | 0.01 | 0.24 | 0.05 | 1.00 |
| 0.10 | 0.10 | 1.50 | 0.39 | 1.03 |
| 1.00 | 1.20 | 22.10 | 4.97 | 2.00 |

**Table 5.** Conductance estimation parameters for $C_x = 100$ pF, $C_c = 100$ pF, and 5 mV rms noise.

| Nominal $G_x$ [PF] | Estimated $G_x$ [PF] | Relative Error [%] | Uncertainty [%] | Sensitivity |
|---|---|---|---|---|
| 0.01 | 0.01 | 0.78 | 0.10 | 1.03 |
| 0.10 | 0.10 | 3.72 | 0.41 | 1.04 |
| 1.00 | 1.50 | 52.20 | 5.02 | 2.00 |

*3.2. Image Reconstruction from Simulated Data*

Two sets of thorax phantom tomographic data were used for the comparison. The first set was generated in a numerical simulation describing the complex capacitance of the object itself without taking into account the admittance of the electrodes or leads. These data, the so-called unobservable data, were generated without adding noise. The second dataset contained measurements simulated by the proposed method, taking into account the coupling capacitance of the measuring electrodes and the noise of the measuring electronics. Additive Gaussian noise was added to the generated digital samples with a magnitude of 5 mV rms.

The capacitance and conductivity measurements simulated for a 32-electrode sensor are shown in Figure 14. For each excitation electrode, 31 measurements can be made using the sensing electrodes. Only 496 measurements, not being a linear combination, were used for the reconstruction. The figure shows only a part of the simulated results for the readability of the charts. The measurement data arranged according to the excitation electrodes form a characteristic graph with a repeating U-shape. In this U-shape graph, the maximum values correspond to measurements from the adjacent electrodes, and the minimum values correspond to measurements with pairs of opposite electrodes. The values of real capacitance and conductance of the object are plotted to illustrate the performance of the proposed method. The conductance values obtained using the proposed method fit well with the real values in almost the whole measurement range. Small discrepancies can be observed for high conductance values, which is related to the relative error at the upper limit of the measurement range (Figure 8). The 5 mV rms noise has a minimal influence on conductance estimation. The impact of noise on capacitance estimation is significant because the capacitance values for the thorax phantom are very close to the lower limit of the measurement range, where the relative standard uncertainty is high (Table 2).

The images of permittivity and conductivity were reconstructed from both datasets: ideal data and data simulated using the numerical model of the proposed method with noise and coupling capacitance. The results of image reconstruction, together with true distributions of permittivity and conductivity in the phantom, are shown in Figure 15.

The normalized L2 norms were calculated to measure the quality of the reconstructed images. The residuum describes how well the algorithm fits the solution to the minimized norm and is given by the formula

$$res\left(x^i\right) = \frac{\|D - J_k x^k\|_2^2}{\|D - J_k x^{true}\|_2^2},$$
(36)

where $D$ is the vector of measurements, $J_k$ is the Jacobian matrix at the $k$-th iteration, $x^k$ is the estimate at the $k$-th iteration, and $x^{true}$ is the true spatial distribution. The L2 distance between the estimate and the phantom describes the discrepancy between the reconstructed and true distributions and is given by

$$dis\left(x^k\right) = \frac{\|x^k - x^0\|_2^2}{\|x^{true} - x^0\|_2^2}$$
(37)

where $x^0$ is the initial distribution in the iterative algorithm. The image quality errors obtained for both datasets are presented in Table 6.

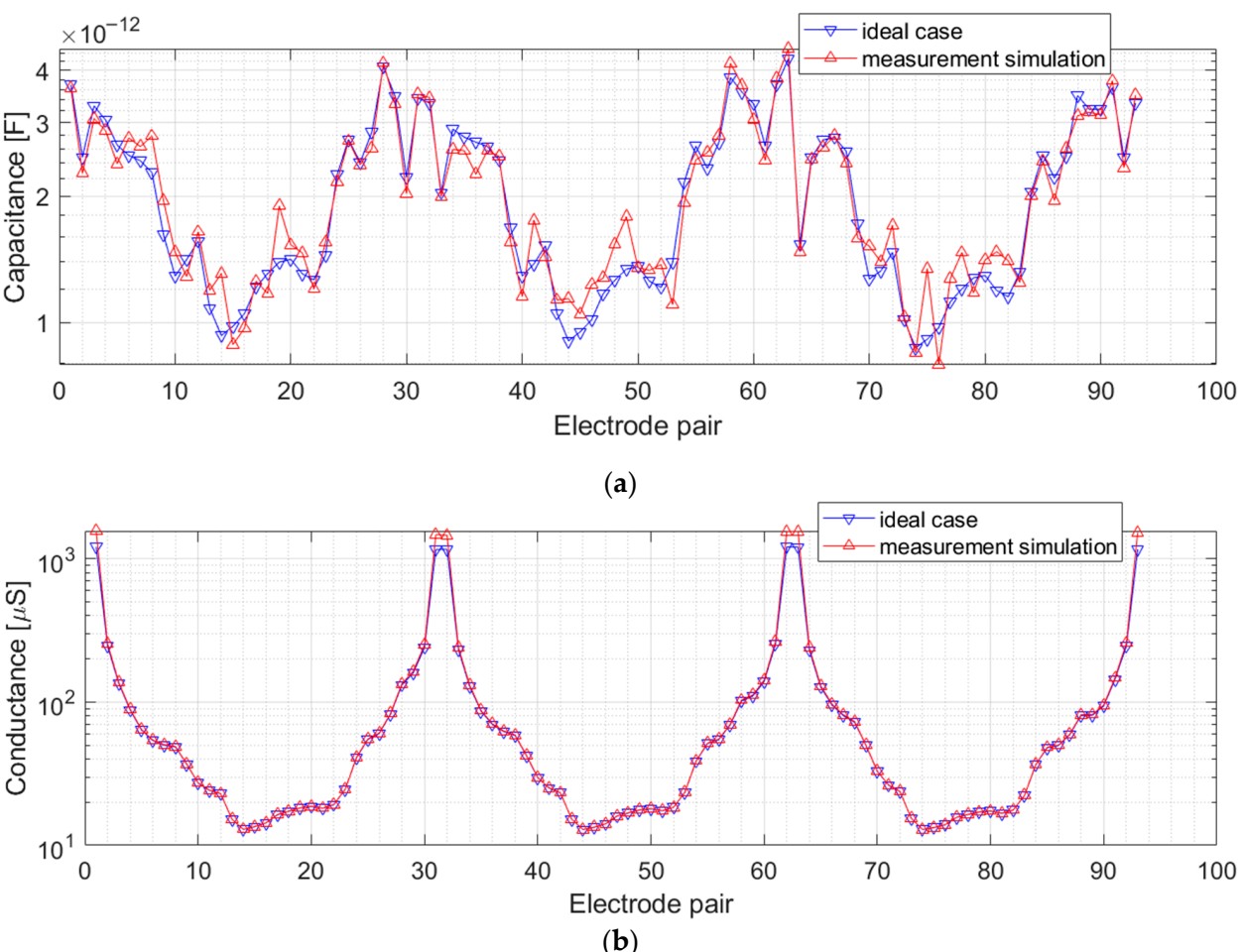

**Figure 14.** Numerical simulated values of capacitance (**a**) and conductance (**b**) for ideal case (blue) and measurement using the pulse excitation (red). Measurements simulated for the thorax phantom. The data for the measurements for three excitation electrodes are shown only.

**Table 6.** Image reconstruction error.

|  | Component | Ideal Data | Data Obtained Using the Proposed Method |
|---|---|---|---|
| Residuum | $\varepsilon$ | $6.6 \times 10^{-4}$ | $3.4 \times 10^{-2}$ |
|  | $\sigma$ | $6.3 \times 10^{-5}$ | $5.2 \times 10^{-4}$ |
| Discrepancy | $\varepsilon$ | 0.125 | 0.389 |
|  | $\sigma$ | 0.580 | 0.711 |

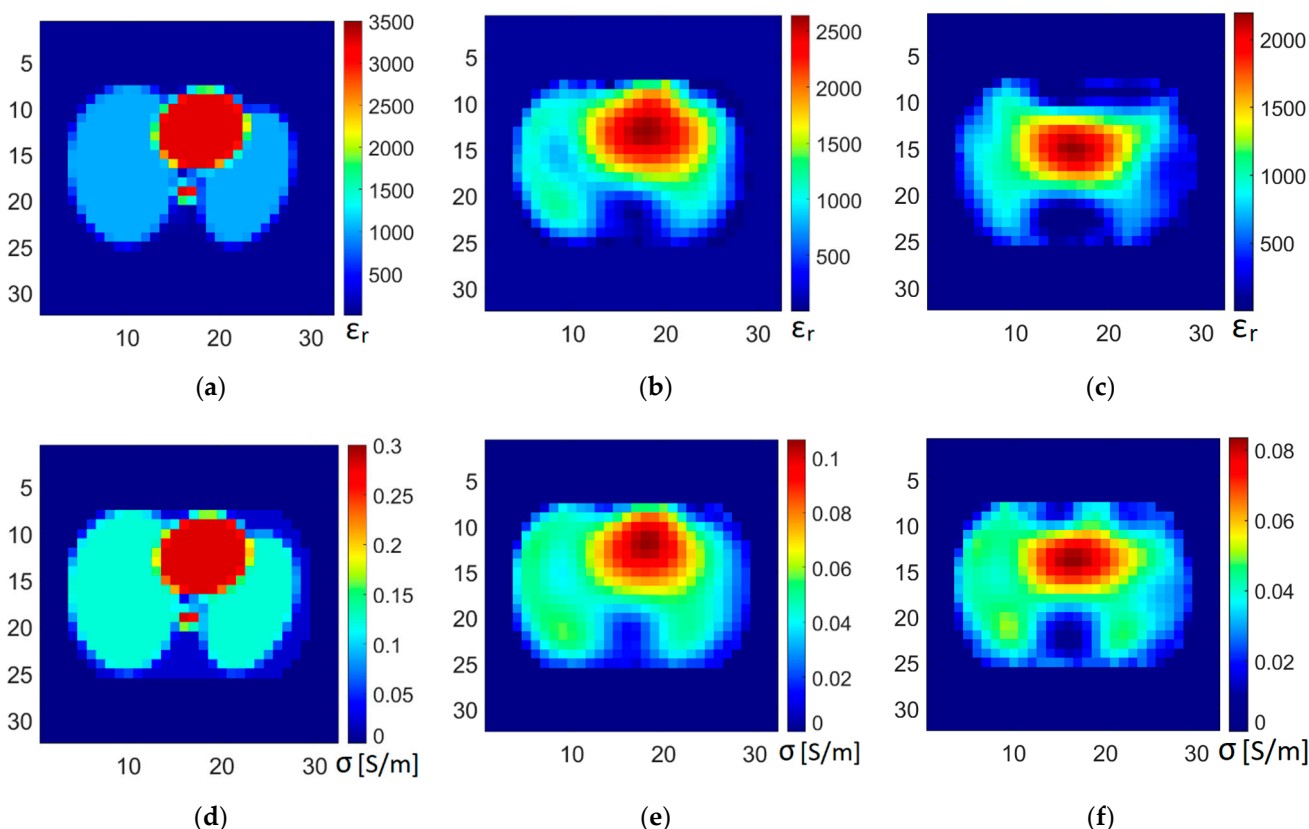

**Figure 15.** Reconstructed images of permittivity and conductivity. True distribution in numerical phantom: (**a**) permittivity, (**d**) conductivity. Images reconstructed from ideal data: (**b**) permittivity, (**e**) conductivity. Images reconstructed from data obtained using the pulse excitation measurement: (**c**) permittivity, (**f**) conductivity.

## 4. Discussion

The paper presents a novel method of capacitance measurement using pulse excitation, which can be an alternative to the commonly used sinusoidal excitation. The advantage of pulse excitation may be the speed of the measurement as well as the broadband stimulation of the examined object. The proposed method uses a simpler two-electrode measurement compared to the four-electrode measurement used in the EIT, where it is necessary to use a current source with high output impedance. The proposed method uses non-contact electrodes, which may be an advantage in the case of medical use due to the patient's comfort and ease of use.

Experiments have demonstrated that on the basis of digital samples of the integrator response, the values of both complex capacitance components can be determined. The performed analysis of the measurement method showed that it could be used in the measuring range from 1 pF to 100 pF for capacitance and from 0.01 mS to 1 mS for conductivity.

The value of the coupling capacitance must be selected so that its susceptance is high in comparison with the measured capacitance and conductivity. Pulse duration and sampling should also suit the measuring range. In the numerical experiment, we used a pulse with a duration of 14 μS and a sampling frequency of 9 MHz. These values made it possible to obtain acceptable measurements and can be easily used in our tomography device in a future hardware implementation of the proposed method. In order to extend the sensitivity range, it is necessary to provide the possibility of changing the pulse width in the measuring system and increasing the sampling frequency. Oversampling will also reduce measurement uncertainty.

In the assumed measurement range, the relative standard uncertainty of measurement, the relative error, and the linearity are acceptable, although the measurement range itself is

limited. The disadvantage of the method is that in case of occurrence of values outside the measuring range, it is necessary to control the data and discard the results.

The conducted numerical experiments have shown that with the proposed measurement method, it is possible to obtain data from which the reconstruction of tomographic images is possible. The reconstructed images do not differ much in quality from the images reconstructed from ideal data (data without noise and the influence of coupling capacitance). For biological tissues, the value of the component related to electric permittivity is much lower than the conductivity component in the range of the frequencies used, which means that the capacitance values are very small compared to the conductivity values. The conductivity images are of better quality than the permittivity images due to the difference in measurement uncertainty for conductivity and permittivity.

## 5. Conclusions

The proposed method of complex capacitance measurement using pulse excitation may be an alternative measurement method for ECT as well as for electrical impedance tomography. It enables imaging of permittivity and conductivity spatial distributions using capacitively coupled electrodes. Thus, it can extend the capability and potential of ECT.

**Author Contributions:** Conceptualization, W.T.S., D.W. and G.D.; methodology, D.W. and W.T.S.; software, O.M., J.K., P.W. and M.M.; validation, G.D. and W.T.S.; formal analysis, W.T.S. and D.W.; investigation, D.W. and O.M.; resources, P.W. and M.M.; data curation, O.M., J.K. and M.M.; writing—original draft preparation, D.W., W.T.S. and O.M.; writing—review and editing, D.W. and W.T.S.; visualization, D.W., O.M. and J.K.; supervision, W.T.S.; project administration, W.T.S.; funding acquisition, W.T.S.; All authors have read and agreed to the published version of the manuscript.

**Funding:** This research was funded by Excellence Initiative—Research University grants by the Ministry of Science and Higher Education (PL), grant number 50404496103445.010402—1820/11/Z01/POB4/2021.

**Institutional Review Board Statement:** Not applicable.

**Informed Consent Statement:** Not applicable.

**Data Availability Statement:** Not applicable.

**Conflicts of Interest:** The authors declare no conflict of interest.

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
