# Peer review of "Numerical Evaluation of Complex Capacitance Measurement Using Pulse Excitation in Electrical Capacitance Tomography"

_electronics, doi:10.3390/electronics11121864_

Round 1
Reviewer 1 Report
In the paper, a method of electrical capacitance tomography (ECT) based on capacitance measurement is proposed. ECT can be potentially used in medicine, similarly to the electrical impedance tomography (EIT). ECT uses capacitively coupled electrodes. Capacitance is measured between the electrodes surrounding the area of interest. The imaging technique is based on measurements of spatial distribution of dielectric permittivity in a domain reconstructed from capacitance values measured between the electrodes surrounding this domain. The proposed method uses a two-electrode measurement. The method enables imaging of spatial distribution of permittivity and conductivity using capacitively coupled electrodes. The numerical experiment was performed. Authors consider the pulse excitation, which is an alternative to the commonly used sinusoidal excitation. The advantage of pulse excitation is a broadband measurement of high speed. The proposed method uses a simple two-electrode measurement of complex capacitance. It enables imaging of permittivity and conductivity spatial distributions using capacitively coupled electrodes. The authors suggested the equivalent circuit for modeling a complex capacitance containing capacitive coupling loaded by integrator-based measurement circuit. The structure is described using the theory of electrical circuits and is presented in details. Pulse shape analysis of the measurement circuit was performed. Response to the rectangular pulse excitation was calculated using Laplace transform for the model of admittance. complex capacitance measurement based on the analysis of the shape of the response. Laplace analysis in case of the rectangular excitation was used to derive the analytical formulas for the components of complex capacitance. The 2D numerical model of the human thorax was used for the numerical experiment. All results obtained are in a good agreement.
The paper contains detailed Conclusions. The References are relevant to the research.
In general, the paper is well written and may be interesting and useful for practical applications.
I recommend to publish the paper in the present form.
Author Response
Dear Reviewer,
Thank you for your insightful review and thorough opinion.
Best regards,
Damian Wanta
Reviewer 2 Report
Authors presented simulations for capacitances up to 100pF, which values are much much higher than these in real measurements. It would be interesting if authors comment this issue.
Author Response
Dear Reviewer,
Thank you for your review.
In the conducted experiments, the circuit parameters corresponded to the current setup of our hardware. These parameters ensure sensitivity and linearity in the range from 1 pF to 100 pF, as was shown in the paper.
Indeed, in the case of the thorax phantom, the capacitance measurements are in the lower part of this range, from 1 to 10 pF. However, the image reconstruction results confirmed that the proposed method can be used for such capacitance values. The size of the electrodes was 15 mm in width and 60 mm in height in the simulation. To increase the capacitance, we can increase the height of the electrodes at the cost of the resolution in the Z-axis.
A wider measuring range (up to 100 pF) of our circuit may be useful in other applications.
Best regards,
Damian Wanta
Reviewer 3 Report
This paper presents a numerical evaluation of complex capacitance measurement using pulse excitation in electrical capacitance tomography.
The work is an interesting approach to complex capacitance measurement methods using pulse excitation. The solution enables the imaging of spatial distributions of permeability and conductivity using capacitively coupled electrodes. The paper's potential application of the presented solution in medical research is of particular note.
I accept the article in its present form.
Author Response
Dear Reviewer,
Thank you for your review.
Best regards,
Damian Wanta